# Thermoreversible Gelation with Supramolecularly Polymerized Cross-Link Junctions

**DOI:** 10.3390/gels9100820

**Published:** 2023-10-15

**Authors:** Fumihiko Tanaka

**Affiliations:** Department of Polymer Chemistry, Graduate School of Engineering, Kyoto University, Katsura, Kyoto 615-8510, Japan; ftanaka@kmj.biglobe.ne.jp

**Keywords:** thermoreversible gelation, supramolecular cross-linking, cooperative polymerization, Bose–Einstein condensation of rings, metal-coordinated supramolecules, ladder junction, egg-box junction

## Abstract

Structure and reversibility of cross-link junctions play pivotal roles in determining the nature of thermoreversible gelation and dynamic mechanical properties of the produced polymer networks. We attempt to theoretically explore new types of sol–gel transitions with mechanical sharpness by allowing cross-links to grow without upper bound. We consider thermoreversible gelation of the primary molecules R{A${}_{f}$} carrying the number *f* of low molecular weight functional groups (gelators) A. Gelators A are assumed to form supramolecular assemblies. Some examples are: telechelic polymers (f=2) carrying π–π stacking benzene derivatives at their both ends, and trifunctional star molecules (f=3) bearing multiple hydrogen-bonding gelators. The sol–gel transition of the primary molecules becomes sharper with the cooperativity parameter of the stepwise linear growth of the cross-links. There is a polymerization transition (crossover without singularity) of the junctions in the postgel region after the gel point is passed. If the gelator A tends to form supramolecular rings competitively with linear chains, there is another phase transition in the deep postgel region where the average molecular weight of the rings becomes infinite (Bose–Einstein condensation of rings). As a typical example of binary cross-links where gelators A and B form mixed junctions, we specifically consider metal-coordinated binding of ligands A by metal ions B. Two types of multi-nuclear supramolecular complexes are studied: (i) linear stacking (ladder) of the sandwich A${}_{2}$B units, and (ii) linear train of egg-box A${}_{4}$B units. To find the strategy towards experimental realization of supramolecular cross-links, the average molecular weight, the gel fraction, the average length of the cross-link junctions are numerically calculated for all of these models as functions of the functionality *f*, the concentration of the solute molecules, and the temperature. Potential candidates for the realization of these new types of thermoreversible gelation are discussed.

## 1. Introduction

Thermoreversible gelation (TRG) in solutions of polymers, as well as of low molecular weight molecules, has been attracting researcher’s interest [1,2,3,4,5,6,7] because of its scientific importance and vast mechanical and biomedical applications of the produced gels. Many examples of the phase diagrams with sol–gel transition lines have been reported in the literature. Some original studies, reviews, and conceptual works have appeared with relation to responsive gels [8,9,10,11,12], hydrogels for biomedical applications [6,7,13], and hydrogen-bonding [14,15,16,17,18,19] and π-functional supramolecular gelators [20,21,22]. The use of weak non-covalent interactions for cross-linking with self-assembly processes in synthetic systems to realize complex multicomponent reversible materials promises possible new attractive functionalities as adhesives, gelators, batteries, anti-fouling coatings, and regenerative medicines. Specific examples of non-covalent interactions utilized are metal–ligand interactions, multiple hydrogen bonding, π-π stacking, host-guest inclusion interactions, and electrostatic interactions.

Most of the studies so far have, however, been concerned on the cross-links of polymers that are confined in small spatial regions. For instance, hydrogen-bonding cross-links are mostly formed by complementary pair of functional groups attached on the primary molecules. Metal-coordinated cross-links are formed by stoichiometric complexes of metal ions and ligands. The cross-linking regions of these interactions are spatially localized in small regions. In contrast, micellar cross-links of hydrophobic short chains, as seen in hydrophobically modified water-soluble polymers [23,24,25,26,27,28,29] (associating polymers), have intermediate size (several tens of hydrophobic groups), but their stable size has an upper limit.

In this paper, we eliminate such restriction on the number of functional groups in a cross-link junction (referred to as cross-link multiplicity *k*), and study TRG with cross-links that can grow without upper bound, such as seen in supramolecular assembly. The specific systems we consider are functional groups (gelators) incorporated within macromolecular structures in several different ways, such as at polymer chain ends, at the termini of the arms of combs/brushes, or within the polymer main chain. They form supramolecular assemblies such as twisted chains (zig-zag array of hydrogen bonds), rings of fibrillar random coils [30,31,32,33], ladders, and egg-boxes. The polymer architecture and number of gelator units per polymer chain (referred to as the functionality *f*) are also adjusted to afford stable supramolecular gels to permit multiple sites of association per polymer chain.

Specific examples of such functional polymers are: hydrogen-bonding polyacrylates with side chains functionalized by ureidopyrimidone (UPy), or adenine-thymine functionalised polymethacrylate co-polymerised with polybutacrylate [32,33], telechelic polysiloxanes endcapped with UPy used as an adhesive, or telechelic poly(isobutylene) with aminoacid residues used [34], and telechelic macromonomers forming metal-ligand supramolecular complexes [35,36,37,38]. A combination of the conventional covalent bonding with macrocycle-based host-guest interactions [39] is another powerful method to realize supramolecular polymer networks.

## 2. Theoretical Method

The model solution we consider is an associating solution in which the number *N* of reactive (associative) molecules (denoted by R{A${}_{f}$}) with degree of polymerization *n* are dissolved in the number $N_{0}$ of solvent molecules (S). We refer to the solution as R{A${}_{f}$}/S. Molecules can be any type, such as high molecular weight linear polymers, star polymers, or low molecular weight polyfunctional molecules, etc. Each molecule carries the number *f* of functional groups A, which can form interchain cross-links made up of variable number *k* of A-groups (multiplicity *k*) [4,40,41,42].

In this paper, we specifically consider low-mass gelators as the functional groups A which are capable of forming supramolecular assembly without upper bound in the multiplicity *k*. Some examples of such reactive molecules are telechelic polymers (f=2) carrying multiple hydrogen-bonding gelators (oil gelators) [32,33], or carrying π–π stacking benzene derivatives [20], at their both chain ends, trifunctional star molecules (f=3) bearing multiple hydrogen-bonding gelators at their arm ends [14,21]. In the solutions of such reactive molecules, self-association of functional groups A takes place.

In contrast to such self-association, we can consider supramolecular assembly consisting of complementary functional groups A and B. Gelation phenomena in such solutions with mixed cross-link junctions can be observed in the mixed solutions R{A${}_{f}$}/R{B${}_{g}$}/S. To study the nature of TRG with supramolecular binary cross-link junctions, we consider metal-coordinated binding of ligands A by metal ions B. The functionality of a metal ion is regarded as g=1. We study two types of multi-nuclear coordinate complexes with metal ions: (i) linear stacking (ladders) of sandwich units A${}_{2}$B, and (ii) linear trains of egg-box units A${}_{4}$B.

### 2.1. Self-Association

Let us start from the self-association. This is based on the lattice-theoretical picture of polymer solutions [43,44], and divides the system volume *V* into cells of size *a* of the solvent molecule, each of which is assumed to accommodate a statistical repeat unit of the reactive molecules. The volume of a reactive molecule is then given by *n*, and that of a solvent molecule is $n_{0}\equiv 1$ in the unit of the cell volume. We assume incompressibility of the solution, so that we have $\Omega =nN+N_{0}$ for the total volume. The volume fraction of each component is then given by ϕ=nN/Ω for the reactive molecule, and $\phi_{0}=N_{0}/\Omega$ for the solvent. In terms of the functional groups, the number concentration of A-groups on the reactive molecules is ψ=fϕ/n.

In our previous work [42,45], we studied TRG and phase separation in solutions of functional molecules with unary (self) cross-linking. We started from the equilibrium condition
(1)$$\frac{n_{k}}{{n_{1}}^{k}}=K_{k}\left(T\right)$$
for the number concentration $n_{k}$ of the cross-link junctions of multiplicity *k*. Here, $K_{k}\left(T\right)$ is the equilibrium constant of the cross-linking reaction, and $n_{1}$ is the concentration of the free A groups. Let $p_{k}$ be the probability for an arbitrarily chosen A group to belong to a cross-link junction of multiplicity *k* (conventionally referred to as equilibrium conversion). Then, we have the relation
(2)$$\psi p_{k}=kn_{k}$$
because there are *k* of A groups in a *k*-junction. The equilibrium condition leads to the relation
(3)$$\psi p_{k}=kK_{k}z^{k}$$
for the reactivity given in terms of the number concentration of the free groups $z\equiv \psi p_{1}$. From the normalization condition of $p_{k}$, we find the conservation law
(4)ψ=zu(z)
where
(5)$$u\left(z\right)\equiv \sum_{k\geq 1}kK_{k}z^{k-1}$$
is a function for the characterization of the cross-linking.

In what follows, we assume, as in the classical theory of gelation [46,47,48,49,50,51], is that (i) all functional groups A are equally reactive (principle of equal reactivity), and (ii) three-dimensional cross-linked polymers take a tree structure; there is no cyclic structure (tree statistics). However, the restriction of covalent pairwise reaction is eliminated so that we can treat arbitrary multiplicity *k* with the conversion $p_{k}$ given by (Equation 3) in terms of the equilibrium constants [40,41,42].

To study TRG with such multiple cross-links, we go back to Good’s theory [52,53,54] of cascade processes, and introduce the probability generating function (p.g.f.)
(6)$$\tilde{W}\left(\theta \right)\equiv \sum_{m\geq 1}W_{m}\theta^{m}$$
where $W_{m}$ is the molecular weight distribution of the cross-linked polymers (*m*-mers), and θ is a mathematical dummy index to transform it to p.g.f. We then apply cascade analysis of the branching processes [52], and find the recursion equations
(7a)$$\begin{array}{ccc} \tilde{W}\left(\theta \right) & = & \theta \tilde{u}{\left(x\right)}^{f} \end{array}$$
(7b)$$\begin{array}{ccc} x & = & \theta \tilde{u}{\left(x\right)}^{f-1} \end{array}$$
for the tree structure, where *x* is the probability for an arbitrarily chosen unreacted functional group to belong to the sol part. It is referred to as *extinction probability* in the cascade theory, because it means the probability that any reacted path starting from an unreacted functional group A does not continue to infinity. The cascade function $\tilde{u}\left(x\right)$ is defined by
(8)$$\tilde{u}\left(x\right)\equiv \sum_{k\geq 1}p_{k}x^{k-1}$$

For TRG for which equilibrium condition (Equation 3) holds, we have
(9)$$\tilde{u}\left(x\right)=\frac{1}{\psi x}\sum_{k\geq 1}kK_{k}{\left(xz\right)}^{k}=\frac{z}{\psi }u\left(xz\right)=\frac{u(xz)}{u(z)}$$
for the cascade function written in terms of the function u(z) for the description of the conservation law. In the pregel region, we have x=1 by definition.

On the basis of these cascade equations, we calculate the weight-average molecular weight ${\bar{M}}_{w}$ measured in terms of the molecular weight *M* of the primary molecule [42,45], and find that in the pregel region it is given by
(10)$$P_{w}\equiv \frac{{\bar{M}}_{w}}{M}=\frac{1+\kappa (z)}{1-f^{\prime }\kappa \left(z\right)}$$
where $f^{\prime }\equiv f-1$, and
(11)$$\kappa \left(z\right)\equiv \sum_{k\geq 2}\left(k-1\right)p_{k}=\frac{d\ln u(z)}{d\ln z}$$
is the average branching number of the cross-links. Hence, for the gel point where ${\bar{M}}_{w}$ diverges, we have the condition
(12)$$D\left(z\right)\equiv 1-f^{\prime }\kappa \left(z\right)=0$$

The average branching number is related to the average multiplicity defined by
(13)$${\bar{\mu }}_{w}\equiv \sum_{k\geq 1}kp_{k}$$
through the relation
(14)$$\kappa \left(z\right)={\bar{\mu }}_{w}\left(z\right)-1$$

(For counting the number of reacted paths going out from a cross-link junction, one path coming into it must be subtracted.)

In the postgel region where the gel point is passed, we must go back to the cascade recursion relation ([Disp-formula FD7b-gels-09-00820]) of the branching process. For the dummy parameter of p.g.f. θ=1, it is an equation
(15)$$x=\tilde{u}{\left(x\right)}^{f^{\prime }}$$

A detailed discussion of this equation is given in the papers by Gordon [52] and Good [53,54]. Fukui and Yamabe [40] also derived the same equation by applying the method of steepest descent to find the molecular weight distribution in the postgel region from p.g.f. For the pairwise reaction as seen in covalent cross-linking, this equation is reduced to Flory’s postgel treatment. For TRG, the equation to find the extinction probability *x* can be transformed to
(16)$$H\left(x\right)\equiv x^{1/f^{\prime }}u\left(z\right)-u\left(xz\right)=0$$

It has a solution $x_{1}\;\left(0<x_{1}<1\right)$ apart from the trivial solution x=1. Because $x_{1}$ has the physical meaning of the probability for an arbitrarily chosen unreacted (free) A group to belong to the sol part, the weight fraction of the sol part $W_{\mathrm{sol}}=\tilde{W}\left(\theta =1\right)$ is given by
(17)$$W_{\mathrm{sol}}=\tilde{W}\left(\theta =1\right)=x_{1}\tilde{u}\left(x_{1}\right)={x_{1}}^{f/f^{\prime }}$$
from the first Equation ([Disp-formula FD7a-gels-09-00820]). Then, the gel fraction is given by
(18)$$W_{\mathrm{gel}}=1-W_{\mathrm{sol}}=1-{x_{1}}^{f/f^{\prime }}$$

Similarly, the weight-average molecular weight of the sol part is found to be
(19)$$P_{w}^{\left(s\right)}=\frac{1+\kappa (x_{1}z)}{1-f^{\prime }\kappa \left(x_{1}z\right)}$$

Therefore, in the postgel region, we have only to replace *z* by $x_{1}z$ to find the average quantities referring to the sol part. While the total average multiplicity of the cross-link junctions is
(20)$${\bar{\mu }}_{w}=\kappa \left(z\right)+1$$
by definition, the average multiplicity of cross-link junctions in the sol part is
(21)$${\bar{\mu }}_{w}^{\left(s\right)}=\kappa \left(x_{1}z\right)+1$$

To summarize, the conservation law (Equation 4), the gel-point condition (Equation 12) and the equation for the extinction (Equation 16) serve as a complete set for the study of TRG with unary cross-linking as functions of the given concentration, temperature, and functionality.

Some examples of the supramolecular cross-linking are shown in Figure 1. In Figure 1a, cross-linked networks consisting of low molecular weight trifunctional (f=3) molecules are shown. Functional groups (low-mass gelators) on a molecule are assumed to form either linear chains or rings of arbitrary length. The multiplicity *k* of a cross-link junction is therefore equivalent to the length of chains and rings. In order to apply the conventional tree statistics (cascade theory) for the study of gelation, we assume all networks take the tree form without forming cycles. Rings considered here are, therefore, not the network cycles, but expanded branch points (branch zones). The smallest ring consists of three reacted functional groups. The molecules bearing more than one reacted functional groups in a network serve as branch points [55].

In Figure 1b, networks consisting of telechelic polymers (n>>1) carrying gelators at their both ends (f=2) are shown. Gelators on a molecule are assumed to form either linear chains or rings of arbitrary length as in Figure 1a. Although the physical properties of the formed gels are very different from those of low-mass trifunctional molecules, the nature of TRG can be studied from a unified theoretical scheme by properly tuning the functionality *f* and the molecular weight *n*.

### 2.2. Linear Growth of the Cross-Link Junctions

Let us first consider the simplest case of stepwise linear growth without rings. The association of A groups starts from the nucleation process
(22)$$J\left(1\right)+J\left(1\right)⇌J\left(2\right)\;:\;\frac{n_{2}}{{n_{1}}^{2}}=\lambda_{2}$$
where a symbol J(k) means a junction of multiplicity *k*, $n_{k}$ is their number concentration, and $\lambda_{2}$ is the association constant of the dimerization. The following step is the repetition of
(23)$$J\left(k-1\right)+J\left(1\right)⇌J\left(k\right)\;:\;\frac{n_{k}}{n_{1}n_{k-1}}=\lambda_{k}\;\left(k=3,4,\cdots \right)$$
with the equilibrium constant $\lambda_{k}$ of the *k*-th step. The total equilibrium constant is then given by
(24)$$K_{k}=\lambda_{2}\lambda_{3}\cdots \lambda_{k}$$

In the special case where all stepwise constants are the same (called *isodemic* association [32]), it is simply
(25)$$K_{k}=\lambda^{k-1}$$

We have already studied TRG and phase separation with such isodemic cross-linking in detail [41]. In the *cooperative* association, we assume the nucleation process requires highly restricted conditions leading to a small equilibrium constant $\lambda_{2}$ compared to the all subsequent steps. The simplest model $\lambda_{2}=\sigma \lambda$ with all other constants $\lambda_{k}$ equal to λ has been extensively studied [30,31,32]. We then have
(26)$$K_{k}=\sigma \lambda^{k-1}$$
with small constant σ (referred to as *cooperativity parameter*). (For σ larger than 1, the model is referred to as *anti-cooperative* association [32].)

This cooperative model with two constants λ and σ can be extended to include variable size *s* of the nucleus, such as
(27)$$\lambda_{k}=\sigma \lambda \;\left(k=2,\cdots ,s-1\right),\;\lambda_{k}=\lambda \;\left(s\geq k\right)$$

Also, we can extend this model to the cross-links for which the *s*-th step is very difficult to go through compared to others. We then have the equilibrium constants
(28)$$\lambda_{k}=\lambda \;\left(k\neq s\right),\;\lambda_{s}=\sigma \lambda$$
for such a *bottle-neck* model. This model may be applied to the *chelate effect* as seen in metal-coordinated complex formation.

For the cooperative growth of linear assembly, we have
(29)$$u\left(z\right)=1+u_{C}\left(\lambda z\right)$$
where the function $u_{C}\left(z\right)$ is defined by
(30)$$u_{C}\left(z\right)\equiv \sigma \sum_{k\geq 2}kz^{k-1}=\sigma \frac{z(2-z)}{{\left(1-z\right)}^{2}}$$

Since the concentration *z* is always scaled by the factor λ, in what follows we write λz as *z*. The conservation law then takes the form
(31)a=zu(z)
where
(32)$$u\left(z\right)=1+u_{C}\left(z\right)=\frac{1-2\left(1-\sigma \right)z+\left(1-\sigma \right)z^{2}}{{\left(1-z\right)}^{2}}$$
and
(33)$$a\equiv \frac{\lambda (T)f}{n}\phi$$
is the scaled concentration of the primary molecules. Because the equilibrium constant λ depends on the temperature, we have explicitly indicated its temperature dependence. Therefore, as far as TRG is concerned, the concentration and temperature always appear as a single combined variable λ(T)ϕ.

Simple differentiation leads to the average branching number
(34)$$\kappa \left(z\right)=\frac{2\sigma z}{\left(1-z\right)[1-2\left(1-\sigma \right)z+\left(1-\sigma \right)z^{2}]}$$

Its proportionality to the parameter σ results in a sharp sol–gel transition of a cooperative chain growth.

To see the nature of TRG with cross-links of supramolecular chain growth, we first numerically solve the three fundamental coupled equations described above. The conservation law (Equation 31) takes the form
(35)$$F\left(z\right)\equiv a{\left(1-z\right)}^{2}-z\left\{1-2\left(1-\sigma \right)z+\left(1-\sigma \right)z^{2}\right\}=0$$
from which we can find the concentration z=z(a) of unreacted functional groups as a function of the total concentration *a*. At the gel point, the condition (Equation 12) gives the numerical value of $z=z_{g}$. Together with the conservation law, we find the gel-point concentration (temperature) is given by
(36)$$a_{g}={\left[\frac{f}{n}\lambda \left(T\right)\phi \right]}_{g}=\frac{2f^{\prime }\sigma {z_{g}}^{2}}{{\left(1-z_{g}\right)}^{3}}$$

In the post-gel region, we have to numerically solve extinction (Equation 16) for a given *z*. Because *z* is a function of *a*, we find $x_{1}=x_{1}\left(a\right)$ as a function of the concentration *a*. Then, the gel fraction $W_{\mathrm{gel}}$ is given by (Equation 18). The reciprocal average length of the cross-links ${\bar{\mu }}_{w}^{-1}$ (Equation 13), and the fraction of the reacted functional groups
(37)$$W_{C}=1-z\left(a\right)/a$$
are also calculated.

To capture an entire view of TRG, in Figure 2, we show all of these important observables plotted as functions of the volume fraction of the primary trifunctional low-mass molecules (f=3,n=6) for a given association constant λ=5.0. The cooperativity parameter is fixed at $\sigma =10^{-3}$ as a typical example. We see that the transition region of TRG where $P_{w}$ goes to infinity is very narrow. At the gel-point concentration $\phi =\phi_{g}$, the extinction probability $x_{1}$ deviates from unity, and decreases with the concentration. The average chain length ${\bar{\mu }}_{w}$ increases with the concentration. At a concentration above the gel point, just after the gel point is passed, it increases sharply in a narrow concentration region. This point can be regarded as the polymerization point [30,31], although it is not a true phase transition accompanied by a singularity, but a very sharp crossover change.

To see how TRG depends on the cooperativity of cross-linking, we also plot these properties in Figure 3 by varying the cooperativity parameter. Figure 3a plots the reciprocal weight-average molecular weight $P_{w}^{-1}$ in the pregel region, and that of the sol part ${P_{w}^{\left(s\right)}}^{-1}$ in the postgel region, together with the gel fraction $W_{\mathrm{gel}}$. We can clearly see that TRG becomes sharper and sharper with a decrease in σ (stronger cooperativity). Since the gel fraction rises sharply after the gel point, we expect the dynamic mechanical modulus of the solution goes up sharply at the gel point, leading to easy experimental detection of the transition point. Similarly, Figure 3b plots the reciprocal chain length of the cross-link junctions ${\bar{\mu }}_{w}^{-1}$ together with the gel fraction $W_{\mathrm{gel}}$. We can see that polymerization transition also becomes sharper with a decrease in σ.

To study TRG near the gel point in more detail, let us expand $P_{w}{\left(z\right)}^{-1}$ in the pregel region in powers of the small deviation of $\epsilon \equiv \left(z_{g}-z\right)/z_{g}$. Simple calculation leads to
(38)$$P_{w}{\left(z\right)}^{-1}≃\frac{f^{\prime }\kappa \left(z_{g}\right)}{1+\kappa (z_{g})}\kappa_{2}\left(z_{g}\right)\epsilon +O\left(\epsilon^{2}\right)$$
where
(39)$$\kappa_{2}\left(z\right)\equiv \frac{d\ln\kappa (z)}{d\ln z}$$

At the gel point, we find
(40)$$\kappa_{2}\left(z_{g}\right)=1+\frac{\left(1-\sigma \right){\left(1-z_{g}\right)}^{2}}{f^{\prime }\sigma }≃\frac{1}{\sigma }\;\left(\mathrm{for}\;\sigma <<1\right)$$

Hence, the amplitude of divergence in $P_{w}$ becomes smaller in proportional to σ.

### 2.3. Chain/Ring Supramolecular Cross-Link Junctions

Let us next consider the effect of ring formation. We assume that the functional group A forms either linear chains with equilibrium constants $K_{k}^{\left(C\right)}$, or rings with $K_{k}^{\left(R\right)}$ (see Figure 1a,b). We then have
(41)$$u\left(z\right)=1+u_{C}\left(z\right)+u_{R}\left(z\right)$$
where
(42)$$u_{C}\left(z\right)\equiv \sum_{k\geq 2}kK_{k}^{\left(C\right)}z^{k-1}$$
and
(43)$$u_{R}\left(z\right)\equiv \sum_{k\geq 3}kK_{k}^{\left(R\right)}z^{k-1}$$

(A minimum ring has the size k=3.) The average branching number is then given by
(44)$$\kappa \left(z\right)\equiv \frac{d\ln u(z)}{d\ln z}=W_{C}\left(z\right)\kappa_{C}\left(z\right)+W_{R}\left(z\right)\kappa_{R}\left(z\right)$$
where
(45)$$W_{C}\left(z\right)\equiv \frac{u_{C}\left(z\right)}{u(z)},\;W_{R}\left(z\right)\equiv \frac{u_{R}\left(z\right)}{u(z)}$$
are the weight fraction of chain cross-links and of ring cross-links. Assuming the uniform association constants $\lambda_{C}=\lambda$ and $\lambda_{R}=\mu \lambda$, we have
(46)$$K_{k}^{\left(C\right)}=\sigma_{C}\lambda^{k-1}$$
for the chain growth as above. For the ring formation, we have assumed random growth in contrast to the directional linear growth of chains. If we assume Gaussian chain statistics for the growth, the ring closure probability [56,57,58,59] is proportional to $1/k^{5/2}$. Hence, we have
(47)$$K_{k}^{\left(R\right)}=\sigma_{R}\frac{{\left(\mu \lambda \right)}^{k}}{k^{5/2}}$$

Scaling the variable *z* by λ, we have the conservation law in the form (Equation 31) with
(48)$$u\left(z\right)=1+\sigma_{C}\frac{z(2-z)}{{\left(1-z\right)}^{2}}+\frac{\sigma_{R}}{z}\Phi \left(\mu z;3/2\right)$$
where
(49)$$\Phi \left(z;\alpha \right)\equiv \sum_{k\geq 3}\frac{z^{k}}{k^{\alpha }}$$
is essentially the Truesdell function [60] of order α. (k=1,2 are excluded from the summation.) We then have
(50)$$\kappa_{C}\left(z\right)=\frac{2}{\left(1-z)(2-z\right)}$$
and
(51)$$\kappa_{R}\left(z\right)=\frac{\Phi (\mu z;1/2)}{\Phi (\mu z;3/2)}-1$$

The concentration *z* of the unreacted groups is physically limited to the range 0<z<1 in the case of chain growth, and to the range 0<z<1/μ in the case of ring growth. If μ<1, the function $u_{C}\left(z\right)$ goes to infinity before $u_{R}\left(z\right)$ does. The cross-links are dominated by the chain formation. TRG in such cases is essentially similar to the one we studied above. On the contrary, if μ>1, the function $u_{R}\left(z\right)$ goes to infinity before $u_{C}\left(z\right)$ does, and therefore only the region 0<z<1/μ is physically meaningful. At the upper limit
(52)$$z^{*}\equiv 1/\mu$$
the function Φ(μz;3/2) in (Equation 48) takes a finite value
(53)$$\Phi \left(1;3/2\right)=\zeta \left(3/2\right)-1-\frac{1}{2^{3/2}}=1.258$$
where ζ(3/2)=2.612 is the numerical value of Rieman’s zeta function at 3/2. In what follows, therefore, we focus on the case μ>1.

With increase in the scaled concentration *a*, the concentration of unreacted functional groups *z* takes a unique value as the solution of the conservation law (Equation 31). The system then reaches the gel point $z=z_{g}$ where the gel-point condition
(54)$$D\left(z\right)=1+u_{C}\left(z\right)\left[1-f^{\prime }\kappa_{C}\left(z\right)\right]+u_{R}\left(z\right)\left[1-f^{\prime }\kappa_{R}\left(z\right)\right]=0$$
is fulfilled.

In the postgel region, when *a* reaches a critical value $a^{*}$ given by
(55)$$a^{*}\equiv z^{*}u\left(z^{*}\right)$$
the total concentration of rings of *finite length* is fixed at this value because the function Φ(z;3/2) has a finite value at μz=1 but it goes to infinity above this value. We then have a situation similar to the Bose–Einstein condensation (BEC) of ideal Bose gases [61,62]. The parameter *z* plays a role of the activity of an ideal Bose gas. Above the concentration $a>a^{*}$, the concentration of the chain is fixed at $a_{C}^{*}=z^{*}u_{C}\left(z^{*}\right)$, and that of the finite rings at $a_{R}^{*}=z^{*}u_{R}\left(z^{*}\right)$. Because the summation in $u_{R}\left(z\right)$ does not include the contribution from rings of infinite size k=∞, the remaining part $a-a^{*}$ should be regarded as rings of infinite size. More precisely, for a system of finite particle number *N*, the upper limit of the summation *k* is bound by the total number of functional groups $k_{\max}=fN$. Therefore, the number of rings with $k=k_{\max}$ increases to the order *N* as soon as the concentration *a* exceeds the critical value $a^{*}$, leading to the finite fraction of the infinite rings. Because the activity is fixed at $z=z^{*}$, the fraction of linear chains is given by $W_{C}=a_{C}^{*}/a$, that of finite rings by $W_{R}=a_{R}^{*}/a$. As a result, the fraction of infinite rings by ${a_{R}}_{\infty }^{*}=1-a^{*}/a$.

Figure 4 shows some important physical quantities plotted against the association constant λ(T) for telechelic polymers f=2,n=30. Instead of changing the volume fraction ϕ, we change λ for tuning the scaled concentration *a* to cover a wide range of its value. Changing ϕ with a constant λ is not enough to cover a range for observing BEC of rings. As an example, parameters are fixed at $\sigma_{C}=3.000,\sigma_{R}=0.050,\mu =1.2$, and the concentration is fixed at a constant ϕ=0.2. In the region of small λ (high temperature), we have only the sol part. The chain fraction $W_{C}$ is much larger than the ring fraction $W_{R}$ in this sol region because the former is proportional to $z^{2}$, while the latter is to $z^{3}$. At the gel point, the gel fraction starts to appear and the extinction probability $x_{1}$ deviates from unity. The cross-links are dominated by linear chains in the critical regions.

However, as λ increases (temperature is lowered) in the postgel region, chain fraction $W_{C}$ shows a peak where ring fraction $W_{R}$ starts to increase. Eventually, the solution with mixed sol and gel reaches the BEC point. At this point the fraction of infinite rings ${W_{R}}_{\infty }$ starts to appear. It increases sharply after the BEC point, while chains and finite rings show kinks (discontinuous slopes) and decrease. The average molecular weight of $P_{w}^{\left(s\right)}$ of the sol part stays constant in this region.

## 3. Metallo-Supramolecular Cross-Link Junctions

Let us move to TRG with binary supramolecular cross-linking. To study mixed cross-link junctions, we consider a model polymer solution consisting of two species of reactive molecules, referred to as R{A${}_{f}$}(A molecule) and R{B${}_{g}\}$ (B molecule), in a common solvent S (mostly water), each carrying the number *f* of functional groups A, and *g* of groups B. Let $n_{A}$ be the number of statistical repeat units on an A molecule, and $n_{B}$ on a B molecule. The molecular weights of them are then $M_{A}=M_{0}^{\left(A\right)}n_{A}$ and $M_{B}=M_{0}^{\left(B\right)}n_{B}$, where $M_{0}^{\left(A\right)}$ and $M_{0}^{\left(B\right)}$ are the molecular weights of their statistical repeat units.

Let $N_{\alpha }$ be the number of molecules of the component α in the solution. The volume fraction of each component is then $\phi_{A}=n_{A}N_{A}/\Omega$ for R{A${}_{f}$}, $\phi_{B}=n_{B}N_{B}/\Omega$ for R{B${}_{g}$}, and $\phi_{0}=N_{0}/\Omega$ for the solvent, where $\Omega \equiv n_{A}N_{A}+n_{B}N_{B}+N_{0}$ is the total volume. The number concentration of A groups and B groups are then given by $\psi_{A}=f\phi_{A}/n_{A}$ and $\psi_{B}=g\phi_{B}/n_{B}$.

Let us first briefly review our theoretical scheme for the study of TRG with binary cross-linking [42,45]. For the stepwise reversible formation of the cross-link junctions
(56)$$k_{1}J\left(1,0\right)+k_{2}J\left(0,1\right)⇄J\left(k_{1},k_{2}\right)$$
with the multiplicity type $\left(k_{1},k_{2}\right)$ varied from small ones to larger, we have the equilibrium conditions
(57)$$\frac{\psi_{A}p_{k_{1},k_{2}}/k_{1}}{{\left(\psi_{A}p_{1,0}\right)}^{k_{1}}{\left(\psi_{B}q_{0,1}\right)}^{k_{2}}}=\frac{\psi_{B}q_{k_{1},k_{2}}/k_{2}}{{\left(\psi_{A}p_{1,0}\right)}^{k_{1}}{\left(\psi_{B}q_{0,1}\right)}^{k_{2}}}=K_{k_{1},k_{2}}$$
where $p_{k_{1},k_{2}}$ is the probability for an arbitrarily chosen A group to belong to a junction J$\left(k_{1},k_{2}\right)$, and let $q_{k_{1},k_{2}}$ be that for a B group. They are the counterparts of the conventional reactivity of the functional groups.

We then have
(58a)$$\begin{array}{ccc} p_{k_{1},k_{2}} & = & p_{1,0}k_{1}K_{k_{1},k_{2}}{z_{A}}^{k_{1}-1}{z_{B}}^{k_{2}} \end{array}$$
(58b)$$\begin{array}{ccc} q_{k_{1},k_{2}} & = & q_{0,1}k_{2}K_{k_{1},k_{2}}{z_{A}}^{k_{1}}{z_{B}}^{k_{2}-1} \end{array}$$
where
(59)$$z_{A}\equiv \psi_{A}p_{1,0}\;z_{B}\equiv \psi_{B}q_{0,1}$$
are the concentration of the free functional groups that remain unreacted in the solution. The conservation laws are given by
(60a)$$\begin{array}{ccc} \psi_{A} & = & z_{A}u_{A}\left(z_{A},z_{B}\right) \end{array}$$
(60b)$$\begin{array}{ccc} \psi_{B} & = & z_{B}u_{B}\left(z_{A},z_{B}\right) \end{array}$$
where functions $u_{A},u_{B}$ are defined by
(61a)$$\begin{array}{ccc} u_{A}\left(z_{A},z_{B}\right) & \equiv & \sum_{k_{1}\geq 1,k_{2}\geq 0}k_{1}K_{k_{1},k_{2}}{z_{A}}^{k_{1}-1}{z_{B}}^{k_{2}} \end{array}$$
(61b)$$\begin{array}{ccc} u_{B}\left(z_{A},z_{B}\right) & \equiv & \sum_{k_{1}\geq 0,k_{2}\geq 1}k_{2}K_{k_{1},k_{2}}{z_{A}}^{k_{1}}{z_{B}}^{k_{2}-1} \end{array}$$
in terms of the equilibrium constants. They have physical meanings of the reciprocal unreactivity $u_{A}\left(z_{A},z_{B}\right)=1/p_{1,0},u_{B}\left(z_{A},z_{B}\right)=1/q_{0,1}$. The coupled conservation equations must be solved for the two unknown variables $z_{A},z_{B}$ as functions of the concentration $\psi_{A},\psi_{B}$ given in the preparation stage of the experiments.

In our previous paper [42,45], we derived the weight-average molecular weight of the three-dimensional polymers (clusters) connected by cross-links. Under the simplifying assumption for the molecular weight $M_{0}^{\left(A\right)}=M_{0}^{\left(B\right)}\equiv M_{0}$, the result (Equation (Equation 26) in [45]) of $P_{w}\equiv {\bar{M}}_{w}/M_{0}$ is
(62)$$\begin{array}{ccc} \phi P_{w}=n_{A}\phi_{A}+n_{B}\phi_{B} & + & \frac{1}{D}\left\{{n_{A}}^{2}\psi_{A}\left[\kappa_{A,A}-\left(g-1\right)D_{\kappa }\right]+{n_{B}}^{2}\psi_{B}\left[\kappa_{B,B}-\left(f-1\right)D_{\kappa }\right]\right\}\\ & + & \frac{n_{A}n_{B}}{D}\left(\psi_{A}\kappa_{A,B}+\psi_{B}\kappa_{B,A}\right) \end{array}$$
where $\phi \equiv \phi_{A}+\phi_{B}$ is the total solute volume fraction. Elements of the branching matrix $\hat{\kappa }$ are defined by the logarithmic derivatives
(63)$$\kappa_{\alpha ,\beta }\equiv \frac{\partial \ln u_{\alpha }}{\partial \ln z_{\beta }}$$
and $D_{\kappa }\equiv \kappa_{A,A}\kappa_{B,B}-\kappa_{A,B}\kappa_{B,A}$ is its determinant. The denominator *D* in $P_{w}$ is defined by
(64)$$D\left(z_{A},z_{B}\right)\equiv 1-f^{\prime }\kappa_{A,A}-g^{\prime }\kappa_{B,B}+f^{\prime }g^{\prime }D_{\kappa }$$

It was referred to as Gordon determinant because it was first presented in their cascade theory of gelation [52] for the mixtures of multi-component reactive molecules. Abbreviated notations $f^{\prime }\equiv f-1$ and $g^{\prime }\equiv g-1$ have been used, since they will frequently appear in the following.

At the gel point, the weight average molecular weight goes to infinity, and hence we have
(65)$$D(z_{A},z_{B})=0$$
for a gel to appear. We have $D(z_{A},z_{B})>0$ for the pregel region, and $D(z_{A},z_{B})<0$ for the postgel region. Materials conservation laws ([Disp-formula FD60a-gels-09-00820]) and ([Disp-formula FD60b-gels-09-00820]), together with the gel point condition (Equation 65), leads to the relation between $\psi_{A}$ and $\psi_{B}$, and therefore gives the sol–gel transition line on the ternary phase plane when parameters $z_{A}$ and $z_{B}$ are eliminated in favor of $\phi_{A}$ and $\phi_{B}$.

In the postgel region where the gel point is passed, we have to find the extinction probabilities $x_{1}$ and $y_{1}$, i.e., the probability for an arbitrarily chosen unreacted A, or B, group to belong to the sol part. They are given by the non-trivial solutions of the coupled equations
(66a)$$\begin{array}{ccc} H_{A}\left(x,y\right) & \equiv & x^{1/f^{\prime }}u_{A}\left(z_{A},z_{B}\right)-u_{A}\left(xz_{A},yz_{B}\right)=0 \end{array}$$
(66b)$$\begin{array}{ccc} H_{B}\left(x,y\right) & \equiv & y^{1/g^{\prime }}u_{B}\left(z_{A},z_{B}\right)-u_{B}\left(xz_{A},yz_{B}\right)=0 \end{array}$$

In what follows in this paper, we focus on the metallo-supramolecular cross- linking [35,36,37,38] by assuming that the B molecule is a metal ion. It has functionality g=1, and is of low molecular weight $n_{B}=1$, but can form multiple cross-links. The gel-point condition is simplified to
(67)$$D\left(z_{A},z_{B}\right)=1-f^{\prime }\kappa_{A,A}\left(z_{A},z_{B}\right)=0$$

Obviously, we have only a trivial solution $y_{1}=1$ for *y* because $g^{\prime }=0$.

### 3.1. Ladder Model

The first model of our supramolecular metal-coordinated cross-link junction is a ladder form in which elementary units of the type J(2,1) (bridge or sandwich) are piled up one by one in layered structure (see Figure 5a). The first step is to form a sandwich
(68)$$2J\left(1,0\right)+J\left(0,1\right)⇌J\left(2,1\right)\;:\;\frac{n_{2,1}}{{n_{1,0}}^{2}n_{0,1}}={\lambda_{1}}^{2}$$

Then, subsequent piling steps follow
(69)$$J\left(2k-2,k-1\right)+J\left(2,1\right)⇌J\left(2k,k\right)\;:\;\frac{n_{2k,k}}{n_{2k-2,k-1}n_{2,1}}=\lambda_{2}$$

The multiplicity index of a ladder junction is specified by
(70)$$\left(k_{1},k_{2}\right)=\left(2k,k\right)\;\left(k=1,2,\cdots \right)$$
where *k* is the number of layers, or equivalently of metal ions, in the cross-links. Let $\lambda_{1}$ be the association constant of an A group within a sandwich unit in (Equation 68), and let $\lambda_{2}$ be the binding constant between the adjacent layers in (Equation 69). The equilibrium constant then takes a form
(71)$$K_{k}\equiv K_{2k,k}={\left(\lambda_{1}^{2}\right)}^{k}{\lambda_{2}}^{k-1}=\sigma {\left(\lambda_{1}^{2}\lambda_{2}\right)}^{k}$$
where $\sigma \equiv 1/\lambda_{2}$ plays a role of the cooperativity parameter for ladder formation.

Scaling the concentrations $\psi_{A}$, $z_{A}$ by $\lambda_{1}$, and $\psi_{B}$, $z_{B}$ by $\lambda_{2}$, we find
(72)$$p_{k}\equiv p_{2k,k}=2\mu kz^{k}/a,\;q_{k}\equiv q_{2k,k}=kz^{k}/b$$

Then, the conservation laws are transformed to
(73a)$$\begin{array}{ccc} a & = & z_{A}+2\mu zu\left(z\right) \end{array}$$
(73b)$$\begin{array}{ccc} b & = & z_{B}+zu\left(z\right) \end{array}$$
where $a\equiv \lambda_{1}\psi_{A}$ and $b\equiv \lambda_{2}\psi_{B}$ are the scaled concentrations,
(74)$$z\equiv {z_{A}}^{2}z_{B}$$
is a combined concentration variable, and
(75)$$\mu \equiv \lambda_{1}/\lambda_{2}$$
is the ratio of the intra- and interlayer association constant. The function u(z) is defined by
(76)$$u\left(z\right)\equiv \sum_{k\geq 1}kz^{k-1}=\frac{1}{{\left(1-z\right)}^{2}}$$
as in the unary cross-linking.

Solving these equations for $z_{A}$ and $z_{B}$, and substituting the results into the definition (Equation 74) of the variable *z*, we find a single equation
(77)$$F\left(z\right)\equiv z-{\left\{a-2\mu zu\left(z\right)\right\}}^{2}\left\{b-zu\left(z\right)\right\}=0$$
for *z* for the conservation law.

To find the branching matrix, we take logarithmic derivatives of $u_{A}$ and $u_{B}$. Simple calculation leads to
(78)$$\hat{\kappa }\left(z\right)=zu\left(z\right)\left[\begin{array}{cc} \frac{2\mu }{a}\left\{1+2\kappa \left(z\right)\right\}, & \frac{2\mu }{a}\left\{1+\kappa \left(z\right)\right\}\\ \frac{2}{b}\left\{1+\kappa \left(z\right)\right\}, & \frac{1}{b}\kappa \left(z\right) \end{array}\right]$$
for the $\hat{\kappa }$-matrix with
(79)$$\kappa \left(z\right)\equiv \frac{d\ln u(z)}{d\ln z}=\frac{2z}{1-z}$$

The gel-point condition is then given by
(80)$$D\left(z\right)\equiv 1-\frac{2f^{\prime }\mu }{a}zu\left(z\right)\left\{1+2\kappa \left(z\right)\right\}=0$$

We have numerically solved these equations and constructed phase diagrams showing the sol–gel transition lines on the ternary phase plane. Figure 5b shows an example of low-mass trifunctional molecules (f=3,n=6) cross-linked by metal ions ($g=1,n_{B}=1$) in a solvent. The ratio of the association constants is fixed at μ=1, while λ is changed from curve to curve. The gel region takes a dome shape, whose top indicates the optimal mixing ratio of the solute components.

To see the behavior of TRG across the gel region, let us introduce the solute volume fraction $\phi \equiv \phi_{A}+\phi_{B}$, and the mixing ratio (composition) $u\equiv \phi_{B}/\phi$ of the solute molecules. Then, we have
(81)$$a=a_{1}\phi \left(1-u\right),\;b=b_{1}\phi u$$
where $a_{1}\equiv \mu \lambda f/n_{A}$ and $b_{1}\equiv \lambda g/n_{B}$. For the numerical calculation, we fix ϕ and plot physical properties as functions of the composition *u*.

In the postgel region, the extinction probability for a metal ion is $y_{1}=1$ because its functionality is g=1, and hence unreacted free ions can exist only in the sol part. The extinction probability of a functional group A should satisfy
(82)$$H\left(x\right)\equiv a\left(1-x^{1/f^{\prime }}\right)-2\mu z\left\{u\left(z\right)-xu\left(x^{2}z\right)\right\}=0$$

By using the non-trivial solution $x_{1}$ of this equation, fraction of the sol part is calculated to be
(83)$$W_{\mathrm{sol}}=\left(1-u\right){x_{1}}^{f/f^{\prime }}+u{\tilde{u}}_{B}\left(x_{1},1\right)$$
where
(84)$${\tilde{u}}_{B}\left(x_{1},1\right)=\frac{1}{b}\left\{b-zu\left(z\right)+{x_{1}}^{2}zu\left({x_{1}}^{2}z\right)\right\}$$

The average molecular weight of the clusters in the sol part in the postgel region is given by
(85)$$P_{w}^{\left(s\right)}=P_{w}\left({x_{1}}^{2}z\right)$$
where $P_{w}\left(z\right)$ is given by (Equation 62). The average length of ladders, including both sol and gel part, is calculated by the definition
(86)$${\bar{\mu }}_{w}=\sum_{k\geq 1}kq_{k}=\frac{1}{b}\left\{z_{B}\left(z\right)+zu\left(z\right)\left[1+\kappa \left(z\right)\right]\right\}$$

Figure 6 shows overviews of the reentrant sol–gel–sol transition of the ladder model for low-mass trifunctional molecules with (a) μ=1.0 and (b) $\mu =10^{-4}$. Excess metal ions brings the solution back to a sol phase because of the lack of A groups. The average molecular weight $P_{w}^{-1}$ in the sol region ($u<u_{1},u_{2}<u$), ${P_{w}^{\left(s\right)}}^{-1}$ in the gel region ($u_{1}<u<u_{2}$), and the gel fraction $W_{\mathrm{gel}}$, the extinction probability $x_{1}$ of the functional group A, the average length ${{\bar{\mu }}_{w}}^{-1}$ of the ladder cross-link junctions, are all plotted as functions of the solute composition *u*. We can clearly see that TRG becomes sharper with smaller ratio μ, or an equivalent decrease in the cooperative parameter σ.

In the postgel region between the solute composition $u_{1}$ and $u_{2}$, the fraction of the gel part shows a peak at a certain value of *u*. It is therefore regarded as the optimal ratio for the gel formation. The extinction $x_{1}$ takes a minimum value near (but not exactly at) this optimal gel point. The average length of the ladder junctions also takes a maximum value near this point.

### 3.2. Egg-Box Model

The second model we consider for supramolecular metal-coordinated cross-link junction is an egg-box form [63,64,65] in which elementary units of the type J(4,1) (egg-box) are piled up one by one in layered structure (see Figure 7). The nucleation of a egg-box is the process
(87)$$4J\left(1,0\right)+J\left(0,1\right)⇌J\left(4,1\right)\;:\;\frac{n_{4,1}}{{n_{1,0}}^{4}n_{0,1}}={\lambda_{1}}^{4}$$

Then, subsequent piling processes follow
(88)$$J\left(2k,k-1\right)+J\left(2,1\right)⇌J\left(2k+2,k\right)\;:\;\frac{n_{2k+2,k}}{n_{2k,k-1}n_{2,1}}=\lambda_{2}$$

The multiplicity index of an egg-box junction is then specified by
(89)$$\left(k_{1},k_{2}\right)=\left(2\left(k+1\right),k\right)\;\left(k=1,2,\cdots \right)$$
where *k* is the number of layers (number of metal ions) in a cross-link. Let $\lambda_{1}$ be the association constant of an A group within an egg-box unit in (Equation 87), and let $\lambda_{2}$ be the binding constant between the adjacent layers in (Equation 88). The equilibrium constant then takes a form
(90)$$K_{k}\equiv K_{2(k+1),k}={\lambda_{1}}^{4}{\left({\lambda_{1}}^{2}\lambda_{2}\right)}^{k-1}=\sigma {\left(\lambda_{1}^{2}\lambda_{2}\right)}^{k}$$
where $\sigma \equiv {\lambda_{1}}^{2}/\lambda_{2}$ plays a role of the cooperativity parameter for the egg-box formation. The reactivities are then given by
(91a)$$\begin{array}{ccc} \psi_{A}p_{k} & \equiv & \psi_{A}p_{2(k+1),k}=2\left(k+1\right)K_{k}{z_{A}}^{2}{\left({z_{A}}^{2}z_{B}\right)}^{k} \end{array}$$
(91b)$$\begin{array}{ccc} \psi_{B}q_{k} & \equiv & \psi_{B}q_{2(k+1),k}=kK_{k}{z_{A}}^{2}{\left({z_{A}}^{2}z_{B}\right)}^{k} \end{array}$$

Scaling the concentrations $\psi_{A}$, $z_{A}$ by $\lambda_{1}$, and $\psi_{B}$, $z_{B}$ by $\lambda_{2}$, we find
(92a)$$\begin{array}{ccc} p_{k} & = & 2\mu \left(k+1\right){z_{A}}^{2}z^{k}/a \end{array}$$
(92b)$$\begin{array}{ccc} q_{k} & = & k{z_{A}}^{2}z^{k}/b \end{array}$$
with
(93)$$\mu \equiv \lambda_{1}/\lambda_{2}$$

The conservation laws are transformed to the simple ones
(94a)$$\begin{array}{ccc} a & = & z_{A}\left\{1+2\mu z_{A}zu_{0}\left(z\right)\right\} \end{array}$$
(94b)$$\begin{array}{ccc} b & = & z_{B}\left\{1+{z_{A}}^{4}u_{1}\left(z\right)\right\} \end{array}$$
where
(95)$$z\equiv {z_{A}}^{2}z_{B}$$
again, and *u* functions are defined by
(96a)$$\begin{array}{ccc} u_{0}\left(z\right) & \equiv & \sum_{k\geq 1}\left(k+1\right)z^{k-1}=\frac{2-z}{{\left(1-z\right)}^{2}} \end{array}$$
(96b)$$\begin{array}{ccc} u_{1}\left(z\right) & \equiv & \sum_{k\geq 1}kz^{k-1}=\frac{1}{{\left(1-z\right)}^{2}} \end{array}$$

We can solve the conservation laws for $z_{A},z_{B}$ as functions of *z*. From ([Disp-formula FD94b-gels-09-00820]), we have
(97)$$z_{B}=\frac{b}{1+{z_{A}}^{4}u_{1}\left(z\right)}$$

Substituting into ([Disp-formula FD94a-gels-09-00820]), we find $z_{A}$ satisfies the equation
(98)$$2\mu zu_{0}\left(z\right){z_{A}}^{2}+z_{A}-a=0$$

Hence,
(99)$$z_{A}=z_{A}\left(z\right)\equiv \frac{1}{4\mu zu_{0}\left(z\right)}\left\{\sqrt{1+8a\mu zu_{0}\left(z\right)}-1\right\}$$

By the definition (Equation 95) of *z*, we have a single equation
(100)$$F\left(z\right)\equiv z-\frac{bz_{A}{\left(z\right)}^{2}}{1+{z_{A}\left(z\right)}^{4}u_{1}\left(z\right)}=0$$
to find a solution of *z* as a function of the concentrations a,b.

By partial differentiation of the conservation laws, we have for the branching matrix
(101)$$\hat{\kappa }\left(z\right)=z{z_{A}\left(z\right)}^{2}\left[\begin{array}{cc} \frac{2\mu }{a}u_{0}\left(z\right)\left\{3+2\kappa_{0}\left(z\right)\right\}, & \frac{2\mu }{a}u_{0}\left(z\right)\left\{1+\kappa_{0}\left(z\right)\right\}\\ \frac{2}{b}u_{1}\left(z\right)\left\{2+\kappa_{1}\left(z\right)\right\}, & \frac{1}{b}u_{1}\left(z\right)\kappa_{1}\left(z\right) \end{array}\right]$$
with
(102a)$$\begin{array}{ccc} \kappa_{0}\left(z\right) & \equiv & \frac{d\ln u_{0}\left(z\right)}{d\ln z}=\frac{z(3-z)}{\left(1-z)(2-z\right)} \end{array}$$
(102b)$$\begin{array}{ccc} \kappa_{1}\left(z\right) & \equiv & \frac{d\ln u_{1}\left(z\right)}{d\ln z}=\frac{2}{1-z} \end{array}$$

The gel-point condition is then given by
(103)$$D\left(z\right)\equiv 1-f^{\prime }\frac{2\mu }{a}{z_{A}\left(z\right)}^{2}zu_{0}\left(z\right)\left\{3+2\kappa_{0}\left(z\right)\right\}=0$$

The equation for finding the extinction probability of A groups in the postgel region takes the form
(104)$$H\left(x\right)=a\left(1-x^{1/f^{\prime }}\right)-2\mu z_{A}{\left(z\right)}^{2}z\left\{u_{0}\left(z\right)-x^{3}u_{0}\left(x^{2}z\right)\right\}=0$$

By using the non-trivial solution $x_{1}$ of this equation, the fraction of the sol part is calculated to be
(105)$$W_{\mathrm{sol}}=\left(1-u\right){x_{1}}^{f/f^{\prime }}+u{\tilde{u}}_{B}\left(x_{1},1\right)$$
where
(106)$${\tilde{u}}_{B}\left(x_{1},1\right)=\frac{1+{\left(x_{1}z_{A}\left(z\right)\right)}^{4}u_{1}\left(x_{1}^{2}z\right)}{1+z_{A}{\left(z\right)}^{4}u_{1}\left(z\right)}$$

The average molecular weight of the clusters in the sol part is then given by
(107)$$P_{w}^{\left(s\right)}\left(z\right)=P_{w}\left({x_{1}}^{2}z\right)$$
where $P_{w}\left(z\right)$ is calculated by using (Equation 62). The average length of egg-boxes, including both sol- and gel part, is calculated by the definition as
(108)$${\bar{\mu }}_{w}=\sum_{k\geq 1}kq_{k}=\frac{z\{1+z_{A}{\left(z\right)}^{4}u_{2}\left(z\right)\}}{bz_{A}{\left(z\right)}^{2}}$$
with
(109)$$u_{2}\left(z\right)\equiv \sum_{k\geq 1}k^{2}z^{k-1}=\frac{1+z}{{\left(1-z\right)}^{3}}$$

Figure 8 shows an overview of the reentrant TRG with metallo-supramolecular egg-box cross-link junctions for the different ratio of the association constants: (a) μ=1.0, and (b) $\mu =10^{-4}$. For a fixed λ, the ratio μ plays a role of the cooperativity parameter. We can clearly see that both sol–gel and gel–sol transition become sharper for smaller μ. Though quantitatively different, nature of TRG with egg-box cross-link junctions essentially similar to that with ladder junctions. In the postgel region between the solute composition $u_{1}$ and $u_{2}$, the fraction of the gel part shows a peak at a certain value of *u*. It is therefore regarded as the optimal ratio for the gel formation. The extinction $x_{1}$ takes a minimum value near this optimal gel point. The average length of the egg-box junctions also takes a maximum value near this point.

## 4. Discussion

On the basis of the observed gel points, we can infer the microscopic parameters from macroscopic measurements. For example, Equation (Equation 36) for the chain model results in
(110)$$\ln\phi =\frac{\Delta H-T\Delta S}{k_{B}T}+A\left(f,n,\sigma \right)$$
for the gel-point concentration, because the association constant takes the form
(111)$$\lambda \left(T\right)=\exp\left[-\left(\Delta H-T\Delta S\right)/k_{B}T\right]$$
in terms of the enthalpy ΔH(<0) and entropy ΔS of the binding. The additive part A is a shift constant
(112)$$A\left(f,n,\sigma \right)\equiv \ln\left\{\frac{2n\sigma f^{\prime }{z_{g}}^{2}}{f{\left(1-z_{g}\right)}^{3}}\right\}$$
which depends only on the functionality and the cooperativity parameter. Therefore, from the experimental measurements of the gel-point concentration as a function of the temperature by rheology, for instance, we can obtain the enthalpy of cross-linking as in the conventional Eldridge-Ferry analysis [66,67]. Further, by changing the functionality *f* with other molecular parameters fixed, information on the cooperativity σ can be obtained.

For the ring closure probability, we applied Gaussian chain statistics, and found it proportional to ∼1/$k^{5/2}$ (including the symmetry number). If the piling of gelators does not obey Gaussian statistics but obeys the scaling law due to the excluded volume effect, the ring closure probability is proportional to $1/k^{\tau }$, where τ=νd+γ−1. (d=3 is the space dimensions, ν=0.6 is the Flory’s exponent [44] of the radius of gyration of a chain, and γ=1.13 is the exponent of the total number of self-avoiding random walks [68].) The exponent τ changes from 2.5 to 2.96, but the nature of the functions Φ(z;τ) (Φ(z;τ),Φ(z;τ−1) are finite while Φ(z;τ−2) is infinite at z=1) remains the same, so that the singular behavior of the conservation law remains the same.

As for the metal-coordinated supramolecular cross-linking, we have used the composition *u* of the metal ions. In a usual experiment, however, metal ions are added into the solutions of functional molecules. The number of metal ions relative to the number of functional groups
(113)$$R\equiv \frac{b}{a}=\frac{a_{1}\left(1-u\right)}{b_{1}u}$$
is a more convenient variable to describe the composition of solute molecules [69,70]. All graphs can easily be transformed for this purpose by taking *R* as the horizontal axis.

## 5. Conclusions

We have presented a very broad theoretical framework for the study of thermoreversible gelation with cross-link junctions that can grow without upper limit. The nature of the sol–gel transition with such supramolecularly polymerized cross-link junctions sensitively depends on the structure of the supramolecules and cooperativity in forming them, as characterized by the stepwise association constants. As frequently observed examples, we have presented four fundamental types: (i) linear (zig-zag) array and ring formation in one-component cross-linking, and (ii) ladder complex and egg-box complex in binary cross-linking. For each of them, the nature of its thermoreversible gelation is summarized in a single unified graph in which variations in the important physical quantities are plotted against either the concentration or the temperature. In particular, it is shown that the cooperativity of supramolecular formation plays a crucial role for exhibiting a sharp sol–gel transition.

From the results of the model calculation, the following conclusions can be drawn:(1)*Chain model*: In addition to the sol–gel transition, there occurs a polymerization transition at a certain concentration just after the gel point is passed under a fixed temperature. The transition is not a true phase transition in the sense that it is not accompanied by any singularity in the physical properties. In particular, the average chain length grows to infinity only in the inaccessible limit of complete reaction. However, its variation becomes sharper and sharper with the cooperativity parameter, leading eventually to a singularity at finite reactivity. The increasing sharpness of the sol–gel transition with cooperativity parameter, in particular sharp rise of the gel fraction, makes the experimental detection of the gel point easier.(2)*Chain/ring model*: Under a certain simple condition on the association constants, a new phase transition occurs at a low temperature (large λ) deep in the postgel region, where the average length of rings goes to infinity. There appears a discontinuity in the physical properties at this *condensation* point of rings. The average molecular weight of the cross-linked polymers, the extinction probability, and the gel fraction all stay constant below this temperature. The transition is analogous to the Bose–Einstein condensation of an ideal Bose gas where a finite fraction of particles falls into the condensate of zero momentum.(3)*Ladder model*: A ladder is one of the simplest structures of multi-nuclear metal-coordinated complexes. As a function of the composition *u* of metal ions, there occur two transitions: one from sol to gel at a low value $u_{1}$, and the other from gel back to sol at a higher value $u_{2}$ (reentrant gel–sol transition). In the gel phase between them, there is a composition *u* at which the gel fraction reaches a maximum (optimal gel point). The average length of the ladder increases around this optimal gel point, but is limited within a finite value, and hence there is no polymerization transition. The ratio μ between the intra-layer association constant and the inter-layer one plays a role of the cooperativity parameter. The transitions become sharper with its decrease.(4)*Egg-box model*: Overall variation in physical observables is the same as the ladder model, although there are some quantitative differences. For instance, the gel fraction becomes asymmetric in the postgel region.

The model solutions proposed in this study have obvious advantages in finding the microscopic parameters regarding the cross-linking reaction, such as association constants, cooperativity parameters, and cross-link multiplicity, etc., from macroscopic measurements on the gelation concentration, or temperature. Thus, supramolecular polymerization is incorporated into the conventional framework of the thermoreversible gelation to have a unified picture of polymer chemistry and supramolecular chemistry. We hope detailed experimental data on thermoreversible gelation with supramolecular cross-link junctions as treated here will be reported in the near future.

## Figures and Tables

**Figure 1 gels-09-00820-f001:**
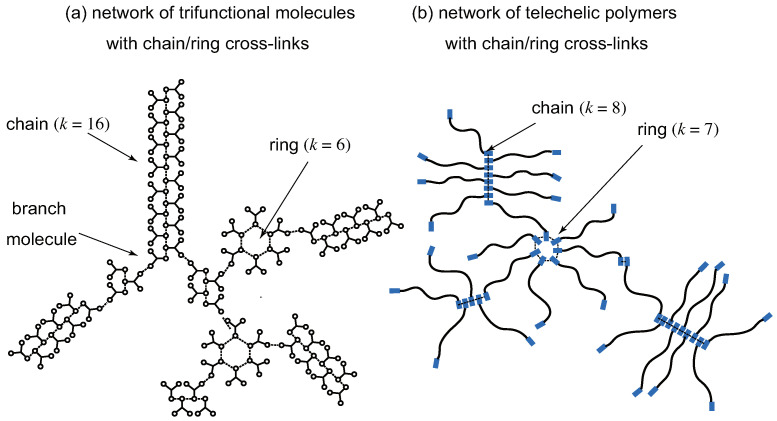
(**a**) A network of a tree type consisting of low molecular weight trifunctional (f=3) molecules with cross-link junctions of linear chains and rings. A chain of the length *k* (dotted line) is regarded as a connected cross-link junction of multiplicity *k*. Similarly, each ring of the length *k* is regarded as a cross-link junction of multiplicity *k* in the loop form. There are branching points where the primary reactive molecules have more than one reacted functional groups. The smallest ring has the size k=3. (**b**) A network consisting of high molecular weight bifunctional (f=2) molecules (telechelic polymers) with coexisting cross-link junctions of linear chains and rings. Functional groups (low-mass gelators) are shown by the blue thick rods at the ends of molecules.

**Figure 2 gels-09-00820-f002:**
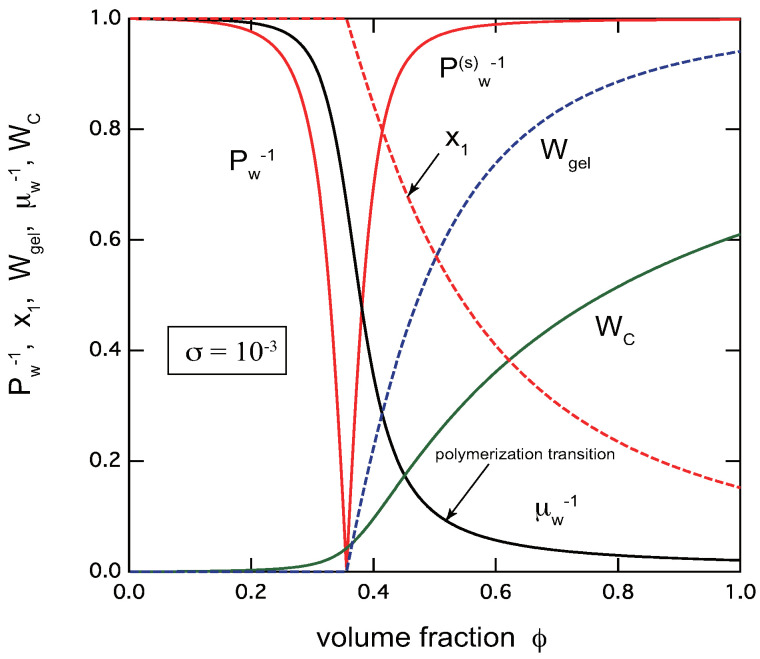
The reciprocal weight-average molecular weight (red solid lines) $P_{w}^{-1}$ in the pregel region, and ${P_{w}^{\left(s\right)}}^{-1}$ in the postgel region, the gel fraction $W_{\mathrm{gel}}$ (blue broken line), the extinction probability $x_{1}$ (red broken line), the reciprocal average chain length ${{\bar{\mu }}_{w}}^{-1}$ (black line), and the fraction $W_{C}$ of the reacted functional groups (green line) plotted against the volume fraction of the primary molecules for f=3,n=6,λ=5.0. The cooperativity parameter is fixed at $\sigma =10^{-3}$. The sol–gel transition is very sharp. There is a polymerization point just after the gel point is passed.

**Figure 3 gels-09-00820-f003:**
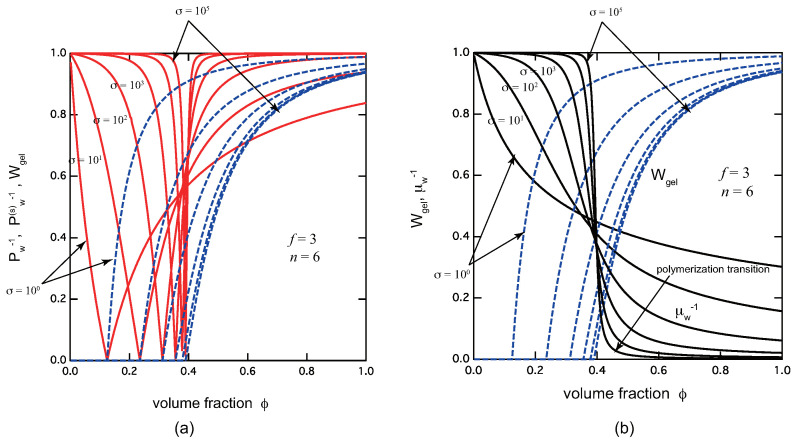
(**a**) The reciprocal weight-average molecular weight (red solid lines) $P_{w}^{-1}$ in the pregel region, and ${P_{w}^{\left(s\right)}}^{-1}$ in the postgel region, and the gel fraction $W_{\mathrm{gel}}$ (blue broken lines) plotted against the volume fraction of the primary molecules. (**b**) The reciprocal average chain length ${\bar{\mu }}_{w}^{-1}$ (black lines), and the gel fraction $W_{\mathrm{gel}}$ (blue broken lines) plotted against the volume fraction of the primary molecules, both for f=3, n=6, λ=5.0. The cooperativity parameter is varied from curve to curve from $\sigma =10^{0}$ to $\sigma =10^{-5}$. Both the sol–gel transition and the polymerization transition become sharper and sharper with decrease in the cooperativity parameter.

**Figure 4 gels-09-00820-f004:**
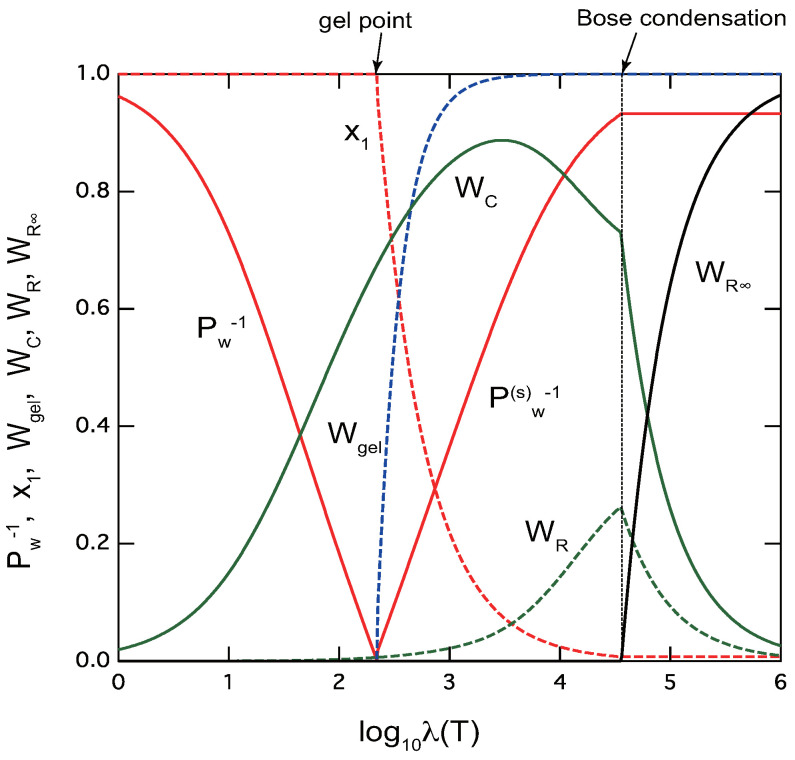
Variation of physical properties characteristic to ring/chain competing TRG of telechelic polymers (f=2,n=30) plotted against the strength λ of the association constant. The reciprocal of the weight-average molecular weight ${P_{w}}^{-1}$ (red line) of the three-dimensional cross-linked polymers in the pregel region, that of the sol parts ${P_{w}^{\left(s\right)}}^{-1}$ (red line) in the postgel region are shown. In the postgel region, we also plot gel fraction $W_{\mathrm{gel}}$ (blue broken line), and extinction probability $x_{1}$ (red broken line). The fraction of chain cross-links $W_{C}$ (green line), and that of ring cross-links $W_{R}$ (green broken line) are plotted in both regions. The fraction of infinite rings ${W_{R}}_{\infty }$ (black line) start to appear at deep point inside the postgel region. The cooperativity parameters are fixed at $\sigma_{C}=3.00,\sigma_{R}=0.05$. In this model calculation, TRG occurs at logλ=2.3, while the second transition (BEC of rings) takes place at logλ=4.6, deep in the postgel region.

**Figure 5 gels-09-00820-f005:**
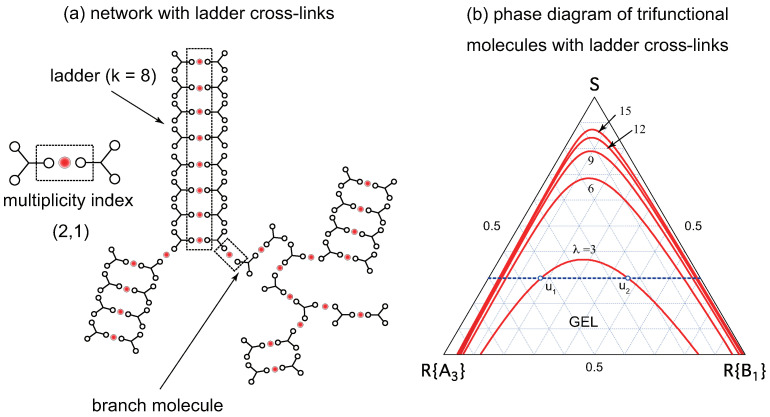
(**a**) Network structure with cross-link junctions of ladder form made up of trifunctional (f=3) low-mass (n=6) molecules. The cross-linker (metal ion) is shown by a red sphere. The elementary unit of a cross-link is a sandwich complex with multiplicty index (2,1). A network is made up of ladder cross-links and branch molecules [55] bearing more than one reacted functional groups. (**b**) Ternary phase diagram for the ladder model of low-mass (n=6) trifunctional (f=3) molecules showing reentrant sol–gel–sol transition (red lines). The association constant λ of the ladder unit is changed from curve to curve at a constant ratio μ=1.0. For a given solute volume fraction ϕ, there are two composition $u_{1}$ and $u_{2}$ for the gel point; the former from sol to gel, and the latter from gel to sol.

**Figure 6 gels-09-00820-f006:**
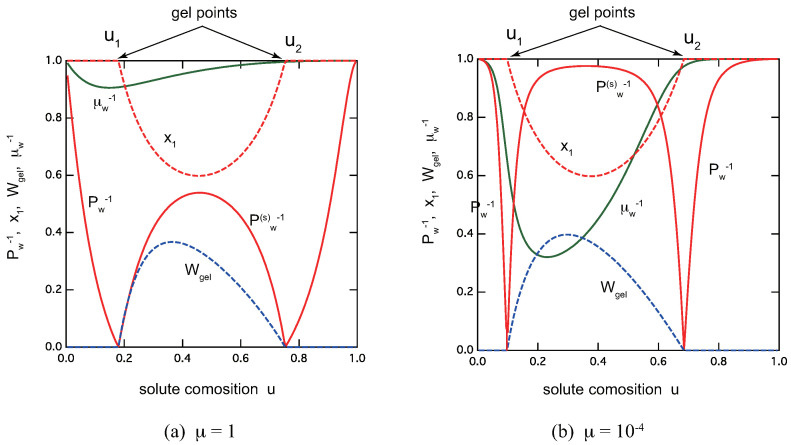
Reentrant TRG with ladder cross-link junctions for trifunctional (f=3) low-mass ($n_{A}=6$) molecules. (**a**) μ=1.0, λ=8.0, (**b**) $\mu =10^{-4},\;\lambda =5.5\times 10^{-3}$. There are a pregel region ($u<u_{1}$), a postgel region ($u_{1}<u<u_{2}$), and a reentrant sol region ($u_{1}<u$). The average molecular weight $P_{w}^{-1}$ in the sol region, ${P_{w}^{\left(s\right)}}^{-1}$ in the gel region, and the gel fraction $W_{\mathrm{gel}}$, the extinction probability $x_{1}$ of the functional group A, the average length ${{\bar{\mu }}_{w}}^{-1}$ of the ladder cross-link junctions, all plotted as functions of the solute composition *u*. The total solute volume fraction is fixed at ϕ=0.3.

**Figure 7 gels-09-00820-f007:**
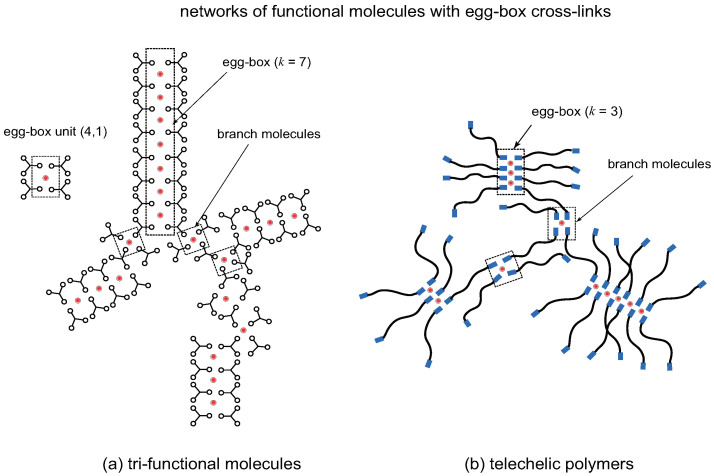
Networks formed by egg-box cross-link junctions made up of (**a**) trifunctional low-mass ($f=3,n_{A}∼1$) molecules, (**b**) telechelic polymers ($f=2,n_{A}>>1$). Cross-linkers (metal ions) are indicated by red spheres. The elementary unit of a cross-link is an egg-box complex with multiplicity index (4,1). A network is made up of linear assembly of egg-boxes and branch molecules bearing more than one reacted functional groups A.

**Figure 8 gels-09-00820-f008:**
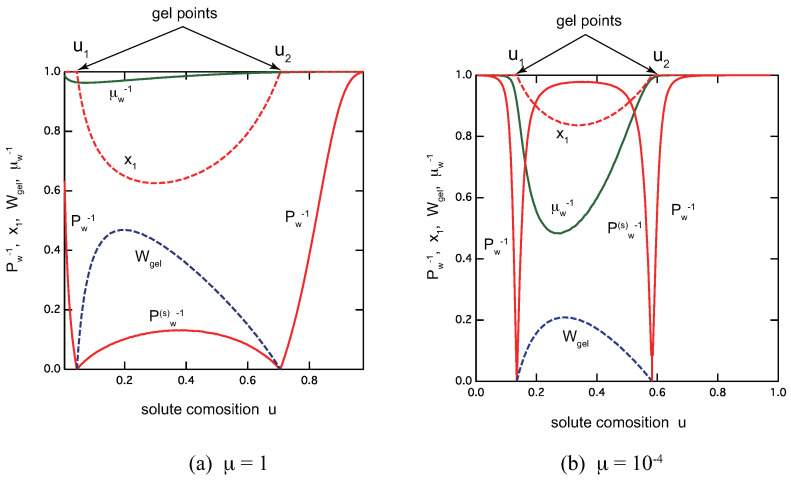
Rentrant TRG with egg-box cross-link junctions for telechelic polymers. (**a**) μ=1.0,λ=40 and (**b**) $\mu =10^{-4},\lambda =1.9\times 10^{3}$. The average molecular weight $P_{w}^{-1}$ in the sol region (red lines), ${P_{w}^{\left(s\right)}}^{-1}$ in the gel region (red line), and the gel fraction $W_{\mathrm{gel}}$ (blue broken line), the extinction probability $x_{1}$ of the functional group A (red broken line), the average length ${{\bar{\mu }}_{w}}^{-1}$ of the egg-box cross-link junctions (green line), all plotted as functions of the solute composition *u*. The total solute volume fraction is fixed at ϕ=0.3.

## Data Availability

The data presented in this study are available within the article.

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
