# Peer review of "Thermoreversible Gelation with Supramolecularly Polymerized Cross-Link Junctions"

_gels, 2023, doi:10.3390/gels9100820_

Round 1
Reviewer 1 Report
The introduction can be improved to show examples of the use of the methodology and better demonstrate the objective of the work. Because there is no results topic, the paper is not totally clear, but the methods are adequately described.
The English is not bad, but there is some minor editing required.
Reviewer 2 Report
This article has presented a very broad theoretical framework for the study of thermoreversible gelation with cross-link junctions that can grow without upper limit. The nature of the sol–gel transition with such supramolecularly polymerized cross-link junctions sensitively depends on the structure of the supramolecules and cooperativity in forming them, as characterized by the stepwise association constants. In particular, it is shown that the cooperativity of supramolecular formation plays a crucial role for exhibiting a sharp sol–gel transition. Therefore, I would like to recommend the publication of the article after minor revision.
Other main issues can be found as follows.
Major points:
1. Some literature examples related to the cross-link junctions can be added to the Introduction part to support the opinion noted in the article.
2. The Figure in the article contains a small number of data figures, such as Figure 3 and Figure 7, which contain only one figure. So, several data figures can be in one figure.
3. There are too many figures in the article. Generally, there are 5 figures in the article. It is suggested to put some data into SI.
4. The author can conduct rheological tests on the gel samples to analyze the influence of different molecules on the gel samples from the perspective of rheology.
5. Regarding the self-assembly of polymer, you can refer to the following literatures: ACS Appl. Mater. Interfaces 2018, 10, 3955−3963
Reviewer 3 Report
The manuscript entitled “Thermoreversible Gelation with Supramolecularly Polymerized Cross-Link Junctions” describes an investigation of new types of sol-gel transitions with mechanical sharpness by allowing crosslinks to grow without upper bounds. The thermoreversible gelation of the primary molecules R{Af} were considered by carrying the number f of low molecular weight functional groups gelators (A). The gelators (i.e., A) are assumed to form supramolecular assemblies. The systems considered are those such as those seen in macromolecular systems such as the ends of polymer chains, the ends of combs or brushes, and within polymer backbone chains. The author adjusted the polymer architecture and the number of gelator units per polymer chain (i.e., functionality, f) to obtain stable supramolecular gels and enable multiple association sites per polymer chain.
This manuscript provides valuable insight about thermoreversible gelation behavior in macromolecular systems and appears to be well-executed. The manuscript is generally well-written, although I have a few suggestions provided below for minor polishing. Although there is a description of the modelling approach used as well as the formulae employed in this investigation, it does not seem to be indicated what type of computational modelling was used in this study (such as what type of software or program was used)? Overall, I believe that this manuscript is suitable for publication pending minor revisions.
Line 61: “twisted chain” can possibly be changed to “twisted chains”.
Line 180: “Although physical properties” can be changed to “Although the physical properties”.
Line 200: “associaition” should be changed to “association”.
Line 240: “This point can be regarded as polymerization point” can be changed to “This point can be regarded as the polymerization point”.
Line 269: For the references, possibly those consecutive references listed as “53,54,55,56” can be listed as “53-56”.
Line 273: “is essentially Truesdell” can be changed to “is essentially a Truesdell” or “is essentially the Truesdell”.
Lines 291-292: A reference citation may be needed for the phrase “We then have a situation similar to the Bose-Einstein condensation (BEC) of ideal Bose gases”.
Line 493: “is more convenient variable” can be changed to “is a more convenient variable”.
Line 524: The phrase “where finite fraction of particles falls into” seems to be unclear.
Lines 546-549: Statements such as the funding and conflicts of interests statements are incomplete.
Overall, the manuscript was well-written and clear. There are a few areas that could benefit from minor polishing, and some suggestions in this regard are provided in my comments to the authors.
